EMBO
Molecular Medicine

# Identification of potential therapeutic targets in prostate cancer through a cross-species approach

Sarah Jurmeister[1,*,†] , Antonio Ramos-Montoya[1,†], Chiranjeevi Sandi[1], Nelma Pértega-Gomes[2], Karan Wadhwa[1], Alastair D Lamb[1,3,4], Mark J Dunning[5] , Jan Attig[6,7] , Jason S Carroll[8], Lee GD Fryer[1], Sérgio L Felisbino[9,†] & David E Neal[1,3,4,†]

## Abstract

Genetically engineered mouse models of cancer can be used to filter genome-wide expression datasets generated from human tumours and to identify gene expression alterations that are functionally important to cancer development and progression. In this study, we have generated RNAseq data from tumours arising in two established mouse models of prostate cancer, PB-Cre/Pten[loxP/loxP] and p53[loxP/loxP]Rb[loxP/loxP], and integrated this with published human prostate cancer expression data to pinpoint cancer-associated gene expression changes that are conserved between the two species. To identify potential therapeutic targets, we then filtered this information for genes that are either known or predicted to be druggable. Using this approach, we revealed a functional role for the kinase MELK as a driver and potential therapeutic target in prostate cancer. We found that MELK expression was required for cell survival, affected the expression of genes associated with prostate cancer progression and was associated with biochemical recurrence.

**Keywords** cross-species analysis; MELK; mouse models; new cancer targets; prostate cancer

**Subject Categories** Cancer; Chromatin, Epigenetics, Genomics & Functional Genomics; Urogenital System

## Introduction

Prostate cancer is the second most frequent cancer in men globally (Torre *et al*, 2015). As the androgen receptor (AR) is the main oncogenic driver in prostate cancer, most drugs used for the treatment of this disease are aimed at inhibiting AR activity (Aragon-Ching, 2014). However, resistance to both first- and second-line androgen deprivation therapies (ADT) commonly occurs (Yuan *et al*, 2014), illustrating the importance of identifying additional therapeutic targets for prostate cancer treatment.

The generation of genome-wide expression datasets from human prostate tumours readily allows the identification of genes whose expression levels are altered during cancer development and progression (Glinsky *et al*, 2004; Taylor *et al*, 2010; Grasso *et al*, 2012; Cancer Genome Atlas Research Network *et al*, 2015; Ross-Adams *et al*, 2015). However, translating this information into clinically useful therapeutic targets poses a twofold challenge: firstly, many genes may be aberrantly expressed as a consequence of cancer development without directly contributing to it, and secondly, not all functionally important genes will represent actionable targets for current drug development approaches.

A number of genetically engineered mouse models (GEMM) of prostate cancer have been developed, and triangulating the findings from genome-wide expression datasets from human cancers with those from well-characterised mouse models may provide a useful filter to prioritise genes that are functionally important to cancer development (Robles-Espinoza & Adams, 2014). Previous studies in various cancer types have demonstrated that integration of gene expression or copy number alteration data from human and murine tumours is a viable approach to derive diagnostic, prognostic or predictive signatures (Belmont *et al*, 2014) and to identify candidate driver genes (Ellwood-Yen *et al*, 2003; Tompkins *et al*, 2013). In prostate cancer, cross-species approaches are particularly

1   Uro-oncology Research Group, CRUK Cambridge Institute, Cambridge, UK
2   Department of Medical Oncology, Dana-Farber Cancer Institute, Harvard Medical School, Boston, MA, USA
3   Department of Urology, University of Cambridge, Cambridge, UK
4   Department of Oncology, Addenbrooke's Hospital, Cambridge, UK
5   Bioinformatics Core Facility, CRUK Cambridge Institute, Cambridge, UK
6   Department of Molecular Neuroscience, UCL Institute of Neurology, London, UK
7   MRC-Laboratory of Molecular Biology, Cambridge, UK
8   Cancer Research UK Cambridge Institute, University of Cambridge, Cambridge, UK
9   Department of Morphology, Institute of Biosciences of Botucatu, Sao Paulo State University (UNESP), Sao Paulo, Brazil
    *Corresponding author. Tel: +44 1223 943698; E-mail: sarah@jurmeister.eu
    †These authors contributed equally to this work

challenging because the mouse prostate is anatomically very different from the human prostate. In contrast to the adult human prostate, the mouse prostate can be divided into histologically distinct lobes: the anterior prostate (AP), ventral prostate (VP), lateral prostate (LP) and dorsal prostate (DP) (Shappell et al, 2004). It has been suggested that the dorsolateral mouse prostate resembles the human peripheral zone (Berquin et al, 2005) where most prostate cancers arise, but this remains a point of debate (Shappell et al, 2004; Irshad & Abate-Shen, 2013). Moreover, to date, no single GEMM by itself is able to faithfully model all aspects of the natural history of prostate cancer (Irshad & Abate-Shen, 2013).

Despite these challenges, two recent publications using such cross-species approaches have reported a synergistic interaction between FOXM1 and CENPE (Aytes et al, 2014) and established a potential role for MET amplifications in prostate cancer (Wanjala et al, 2014), demonstrating the utility of using GEMM as biological filters in the study of this cancer type. However, the full potential of cross-species comparisons to systematically identify and validate druggable targets has not yet been realised. The number of potentially druggable genes in the human genome has been estimated at ~3,000 (Hopkins & Groom, 2002). This represents only 5% of all annotated genes (15% of protein-coding genes), and thus, the majority of genes that are linked to cancer development or progression do not likely make suitable targets for drugs. Taking these considerations into account when prioritising hits from cross-species studies for follow-up validation should thus result in a more focussed identification of potential therapeutic targets.

In this study, we have used next-generation sequencing to obtain detailed gene expression information from different stages of tumour development and progression in two established GEMM of prostate cancer: the PB-Cre/Pten$^{loxP/loxP}$ model, which develops castration-sensitive, invasive but rarely metastasising cancer (Wang et al, 2003; Svensson et al, 2011), and the PB-Cre/p53$^{loxP/loxP}$Rb$^{loxP/loxP}$ model, which develops metastatic castration-resistant prostate cancer (Zhou et al, 2006). Together, these two models represent useful tools to study both indolent and advanced prostate cancer. Furthermore, deletions and mutations of the tumour suppressors Pten, p53 and Rb are among the most common genomic alterations in human prostate cancer, with two-thirds of prostate cancers showing alterations in at least one of the three genes (Appendix Fig S1A).

We have integrated the genomic data obtained from these models with published human prostate cancer expression data and filtered them for potentially druggable genes. Using this approach, we have identified maternal embryonic leucine zipper kinase (MELK) as a potential therapeutic target in prostate cancer. We have then validated the functional importance of MELK for tumour growth both in vitro and in vivo and identified a new mechanism through which this kinase may drive proliferation and viability of prostate cancer cells.

# Results

### Generation of genome-wide expression data representing different stages of murine prostate cancer development and progression

We analysed the histopathology and transcriptomic landscape of prostate tumours arising in all four lobes of the PB-Cre/p53$^{loxP/loxP}$Rb$^{loxP/loxP}$ and PB-Cre/Pten$^{loxP/loxP}$ mouse models of prostate cancer at three different stages of tumour development and progression in order to derive specific signatures for them. A total of 94 samples were selected for further RNAseq analysis, including 20 normal prostatic lobes as well as PIN and tumours from both mouse models (Table EV1). In contrast to a recent publication reporting that high-quality RNA can only be obtained from the lateral and ventral lobes (Zingiryan et al, 2017), we were able to obtain RNA with sufficient RNA quality from all lobes (Table EV1).

In the PB-Cre/Pten$^{loxP/loxP}$ model, all four lobes developed PIN that progressed to invasive adenocarcinoma, albeit with varying kinetics; for example, tumours developing in the AP displayed a delayed progression as compared to the other three lobes, consistent with previous reports (Fig 1A; Wang et al, 2003; Svensson et al, 2011). It is currently unknown whether tumours arising in one lobe resemble human prostate cancer more closely than those arising in others; therefore, we analysed all four lobes separately. Furthermore, to represent different stages of tumour progression, lesions observed in this model were divided into PIN (regions of low-grade and high-grade PIN, predominant in the glands from animals at ages of 3–7 months), medium-stage tumours (regions with high-grade PIN, microinvasive adenocarcinoma and stromal desmoplastic reaction, predominant in prostatic lobes of animals older than 8 months) and advanced-stage tumours (regions with well-differentiated, moderately differentiated and poorly differentiated, frankly invasive adenocarcinoma, predominant in the prostatic lobes of animals older than 12 months) (Fig 1A–D, Appendix Fig S1B). In contrast, PB-Cre/p53$^{loxP/loxP}$Rb$^{loxP/loxP}$ mice developed low-grade PIN in the four lobes (animals older than 4 months), but the aggressive, metastatic tumours that develop in these mice arise from the proximal region of the ducts near the urethra (found in virtually all animals older than 5 months), consistent with previous observations (Zhou et al, 2007). Therefore, periurethral tumours were included in the study in addition to PIN lesions from all four lobes (Fig 1E and F).

### Expression profiling reveals the dynamics of gene expression changes during prostate cancer progression

By selecting samples representing different anatomical regions and stages of tumour progression, our study design provided us with the opportunity to uncover the genetic programmes contributing to prostate cancer development. The pattern of genes expressed in normal mouse prostate lobes was sufficiently distinct for the wild-type samples to be clearly separated by hierarchical clustering according to their lobe of origin (Fig 2A and B). In contrast, in PIN lesions and tumours in the PB-Cre/Pten$^{loxP/loxP}$ model, this distinction was lost (Fig 2A). PIN lesions, medium-stage tumours and advanced-stage tumours were separated from normal samples, but not from each other, although there was a trend for the advanced-stage tumours to cluster together. Conversely, in the PB-Cre/p53$^{loxP/loxP}$Rb$^{loxP/loxP}$ model, PIN and normal samples largely clustered together while tumours were highly distinct (Fig 2B). Taken together, these observations are consistent with the distinct natural histories of tumorigenesis in the two models, with PIN lesions and invasive carcinoma representing a continuum of tumour progression in the PB-Cre/Pten$^{loxP/loxP}$ model, while the PIN lesions and tumours in the PB-Cre/p53$^{loxP/loxP}$Rb$^{loxP/loxP}$ model are distinct entities.

**Figure 1.**

◀

**Figure 1.  Representative histopathological images of the GEMM prostatic lobes.**

A        Wild-type prostate from Cre-negative Pten$^{loxP/loxP}$ mice.

B–D    Stages of cancer development and progression in PB-Cre/Pten$^{loxP/lox}$ mice. PIN: prostatic intraepithelial neoplasia; MedTumour: medium-stage tumour; AdTumour: advanced-stage tumour.

E        Low-grade PIN in PB-Cre/p53$^{loxP/loxP}$Rb$^{loxP/loxP}$ mice (arrows).

F        Prostate tumour arising from the stem/progenitor cell-enriched proximal region of prostatic ducts, shown in two magnifications. AP: anterior prostate; VP: ventral prostate; LP: lateral prostate; DP: dorsal prostate; SV: seminal vesicle; UM: urethral muscle.

Data information: Scale bars correspond to 200 μm, except (F), left: 2 mm.

Due to the slow progression and involvement of all four lobes of the PB-Cre/Pten$^{loxP/loxP}$ model in all stages of tumour progression, we used the data derived from this model to further investigate whether subsets of gene expression alterations could be linked to specific prostate lobes or stages of tumour progression. PIN lesions and tumours arising in the PB-Cre/Pten$^{loxP/loxP}$ model could not be clearly differentiated based on their lobe of origin. This finding suggested that at least some of the gene expression changes that occurred during tumorigenesis were similar between lobes. Indeed, a comparison of genes that were differentially expressed in tumours of the four lobes relative to normal prostate revealed that, while some gene expression changes were unique to a particular lobe, the majority were shared with at least one other lobe (Fig 2C). By selecting the genes that were differentially expressed in all four lobes in a given stage of tumorigenesis, we were thus able to create expression signatures that were associated with PIN lesions, medium-stage tumours (MedTumour) and advanced-stage tumours (AdTumour) in the PB-Cre/Pten$^{loxP/loxP}$ model.

When comparing these three expression signatures of the PB-Cre/Pten$^{loxP/loxP}$ model, we found that there was a core set of 351 genes (262 upregulated and 89 downregulated) that was already aberrantly expressed in PIN lesions and remained dysregulated throughout the subsequent stages of tumour progression (Fig 2D). Progression to medium-stage and advanced-stage tumours resulted in the acquisition of additional gene expression alterations. While some differentially expressed genes (DEGs) were specific to PIN lesions or medium-stage tumours, advanced-stage tumours had the highest number of unique gene expression alterations.

To gain insight into the biological processes associated with tumour progression in the PB-Cre/Pten$^{loxP/loxP}$ model, we performed MetaCore™ enrichment analyses on three different gene sets: genes that were aberrantly expressed in all three stages of tumour progression ("All Stages", $n = 351$); genes that were aberrantly expressed in both medium-stage and advanced-stage tumours, but not in PIN lesions ("Tumour-Specific", $n = 220$); and genes that were only aberrantly expressed in advanced-stage tumours ("AdTumour-Specific", $n = 339$). For each gene set, we ranked all significantly enriched process networks according to the *P*-value of the enrichment and compared these ranks between the three gene sets (Fig 2E). We found that the core set of genes that were altered in all three stages of progression was highly enriched for genes related to cell cycle control. Additional changes in cell cycle-related genes occurred during progression to adenocarcinoma, as shown by the enrichment for these processes in the "Tumour-Specific" gene set, whereas we observed no significant enrichment for these biological processes among genes that were only dysregulated in advanced-stage tumours. Similarly, a significant number of changes in the

expression of cytoskeletal genes occurred during progression both from normal prostate to PIN and from PIN to carcinoma, but not from medium-stage tumours to advanced-stage tumours. Process networks related to inflammation and immune response were enriched among all three gene sets, but the signalling pathways associated with the different stages of progression were largely distinct. Finally, blood vessel morphogenesis was the most enriched process network among "AdTumour-Specificfic" genes, and the enrichment for this biological process was unique to this gene set. Taken together, these analyses reveal the dynamics of gene expression and cellular pathways that are important during the different stages of tumour progression.

**Human and murine prostate tumours share a common set of gene expression alterations**

An important point to consider when interpreting findings obtained from GEMM of cancer is how well the chosen model resembles the human disease. Thus, we next aimed to understand whether the gene expression signatures we derived from tumours arising in mouse models were relevant for the investigation of human prostate cancer. In human prostate cancer, an established and clinically validated cell cycle progression (CCP) score consisting of 31 genes can be used to distinguish indolent from aggressive prostate cancer (Cuzick *et al*, 2011). Examining the mouse homologues of the CCP signature genes, we found that in the PB-Cre/Pten$^{loxP/loxP}$ model, these genes were progressively upregulated during tumour progression and were uniformly highly expressed in the very aggressive PB-Cre/p53$^{loxP/loxP}$Rb$^{loxP/loxP}$ model (Fig 3A). We also assessed a gene expression signature driven by the transcription factor HES6 which has been implicated in prostate cancer progression and resistance to castration (Ramos-Montoya *et al*, 2014). The mouse homologues of these HES6 signature genes were also strongly upregulated in the aggressive, castration-resistant PB-Cre/p53$^{loxP/loxP}$Rb$^{loxP/loxP}$ model, but largely unchanged in the slowly progressing, castration-sensitive PB-Cre/Pten$^{loxP/loxP}$ model (Fig EV1A). This suggested that similar genes may be contributing to aggressiveness in both human and murine prostate tumours.

To further address this question, we used gene set enrichment analysis (GSEA) to compare genes that were aberrantly expressed in murine prostate tumours to a published human prostate cancer dataset (Grasso *et al*, 2012). We found that genes that were deregulated in tumours of both the PB-Cre/Pten$^{loxP/loxP}$ and the PB-Cre/p53$^{loxP/loxP}$Rb$^{loxP/loxP}$ model showed predominantly similar expression patterns in human tumours (Fig 3B). Notably, the strongest enrichments were only obtained by taking into account information from all four murine prostate lobes and both models (Fig EV1B). In

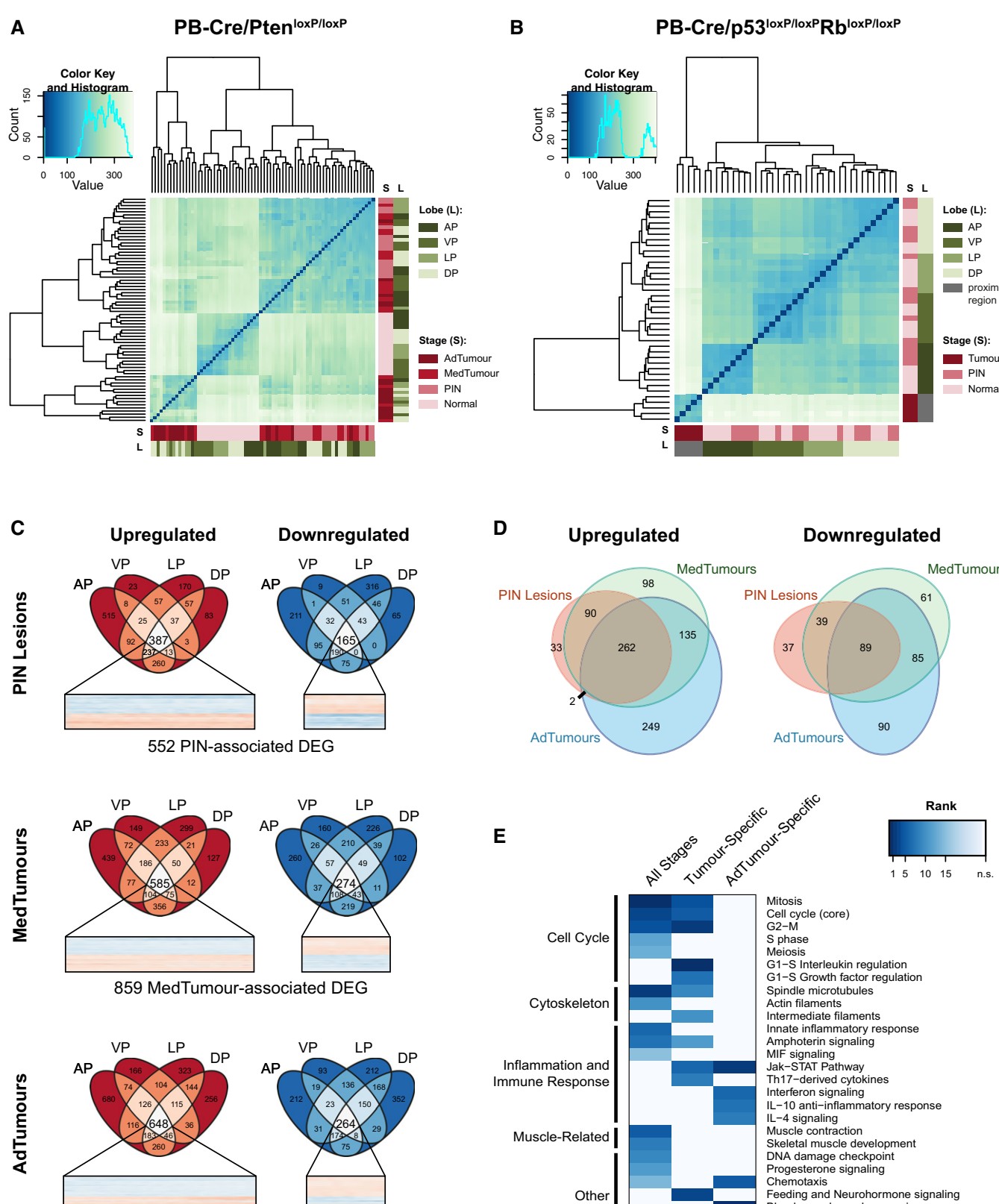

**Figure 2.**

**Figure 2.  Transcriptomic landscape of prostate tumours arising in the PB-Cre/Pten$^{loxP/loxP}$ and PB-Cre/p53$^{loxP/loxP}$Rb$^{loxP/loxP}$ models.**

A    Sample distance heatmap showing the clustering of normal tissue, PIN, medium-stage tumours (MedTumour) and advanced-stage tumours (AdTumour) derived from the four murine prostatic lobes in PB-Cre/Pten$^{loxP/loxP}$ mice based on their gene expression profile as assessed by RNA sequencing.

B    Sample distance heatmap showing the clustering of normal tissue and PIN derived from the four murine prostatic lobes and aggressive, proximal zone-derived tumours in PB-Cre/p53$^{loxP/loxP}$Rb$^{loxP/loxP}$ mice based on their gene expression profile as assessed by RNA sequencing.

C    Venn diagrams showing the derivation of PIN-, MedTumour- and AdTumour-associated gene expression signatures in the PB-Cre/Pten$^{loxP/loxP}$ model. Differentially expressed genes (DEG) in PIN, medium-stage tumours and advanced-stage tumours of all lobes were identified relative to their respective wild-type lobe of origin ($P_{adj}$ < 0.05). Genes that were significantly upregulated or downregulated in PIN, medium-stage tumours or advanced-stage tumours in all four lobes are depicted in heatmaps.

D    Overlaps between the PIN, MedTumour and AdTumour gene expression signatures. PIN: prostatic intraepithelial neoplasia; MedTumour: medium-stage tumour; AdTumour: advanced-stage tumour.

E    MetaCore$^{TM}$ enrichment analyses for process networks on three gene sets: aberrantly expressed in all three stages of tumour progression ("All Stages", n = 351); aberrantly expressed in both medium-stage and advanced-stage tumours, but not in PIN lesions ("Tumour-Specific", n = 220); aberrantly expressed in advanced-stage tumours only ("AdTumour-specific", n = 384). For each gene set, significantly enriched process networks (P < 0.01) were ranked according to their P-value. The lists of enriched process network terms were combined, and their ranks across all three gene sets are shown as a heatmap. n.s.: not significantly enriched in this gene set.

Source data are available online for this figure.

particular, we found that data from individual lobes in the PB-Cre/Pten$^{loxP/loxP}$ model poorly identified genes upregulated in human prostate cancer.

Taken together, our data suggest that human and murine prostate cancers share a common set of gene expression changes and that similar genes contribute to cancer progression in both species.

**Cross-species analysis identifies potential therapeutic targets in prostate cancer**

Having established that murine and human prostate tumours exhibit common gene expression alterations, we hypothesised that integrating information from both species would enable us to enrich for functionally important genes and thus potential novel therapeutic targets. We thus selected all genes that were significantly dysregulated in tumours of both mouse models ("Pten model (all lobes) and p53/Rb model" in Fig EV1B) and in human prostate cancers and filtered them using *The Drug Gene Interaction Database* (DGIdb; http://dgidb.genome.wustl.edu/) to enrich for genes that are predicted to be druggable (Griffith *et al*, 2013; Fig 3C). Furthermore, as most targeted cancer therapies in use today are inhibitors of molecular targets, rather than activators (Abramson, 2014), we chose to focus on genes whose expression was increased in prostate tumours. By using these multiple levels of filtering, we were able to identify 10 potential therapeutic targets (Fig 3D), of which nine were also highly expressed in a second independent prostate cancer dataset (Fig EV2A; Ross-Adams *et al*, 2015). Notably, this group included several key cell cycle regulators (e.g. Bub1, Bub1b, Cdc25c, Cdk1).

To further narrow down this list to the most promising therapeutic targets, we investigated whether the high expression of any of the candidate genes was associated with poor outcome (decreased relapse-free survival) in four published prostate cancer datasets (Glinsky *et al*, 2004; Taylor *et al*, 2010; Ross-Adams *et al*, 2015). Six out of ten genes displayed a statistically significant association with outcome in at least one of the four datasets, suggesting that our approach did indeed uncover a number of genes that are likely to be of functional relevance. In every single one of these cases, high expression of the gene was linked to decreased relapse-free survival (Fig EV2B). One of these genes (MELK) correlated with poor outcome in all datasets (Figs 4A and EV2B and C).

Furthermore, in the Cambridge and Stockholm datasets, MELK is among a list of 100 strongest genes that stratify prostate cancers into distinct molecular subgroups with different clinical outcomes based on integration of copy number and transcript data from radical prostatectomy specimens (Ross-Adams *et al*, 2015). Indeed, the expression of MELK tends to be higher in groups associated with poor prognosis (iCluster 1, iCluster 3 and iCluster 5) than those associated with good prognosis (iCluster 2 and iCluster 4; Figs 4B and EV2D; Ross-Adams *et al*, 2015; Dunning *et al*, 2017). A number of publications suggest that MELK may play an important function in various cancer types, particularly in brain and breast cancers (Marie *et al*, 2008; Hebbard *et al*, 2010; Gu *et al*, 2013; Wang *et al*, 2014). Nevertheless, the molecular mechanisms underlying MELK function in promoting cancer progression remain poorly understood, at least in prostate cancer, and so far, although we have found MELK to be consistently upregulated across multiple prostate cancer datasets (Fig EV2E), only two publications have reported a potential role in prostate cancer (Kuner *et al*, 2013; Ross-Adams *et al*, 2015). Therefore, due to its emerging role as a potential therapeutic target in multiple cancers, as well as the recent development of a small-molecule MELK inhibitor (Chung *et al*, 2012), we selected this kinase for further study in order to determine its value as a potential therapeutic target in prostate cancer.

**MELK is overexpressed in human prostate tumours and associated with aggressiveness**

After first validating the overexpression of MELK mRNA in a second, independent set of PB-Cre/Pten$^{loxP/loxP}$ and the PB-Cre/p53$^{loxP/loxP}$Rb$^{loxP/loxP}$ prostate tumour samples (Fig EV3A), we aimed to confirm that MELK expression is also increased at the protein level in these tumours. We were able to demonstrate increased levels of MELK protein in PIN lesions, medium-stage and advanced-stage tumours in the PB-Cre/Pten$^{loxP/loxP}$ model, and in PIN, primary tumours and liver metastases in the PB-Cre/p53$^{loxP/loxP}$Rb$^{loxP/loxP}$ model by immunohistochemistry (IHC) (Fig EV3B–D).

We then tested whether MELK protein was also overexpressed in human cancers using a tissue microarray. We found that, while the majority (~70%) of non-neoplastic samples were negative for MELK expression, ~60% of PIN samples and more than 80% of tumour samples stained positively (Fig 4C and D). About half of the rare

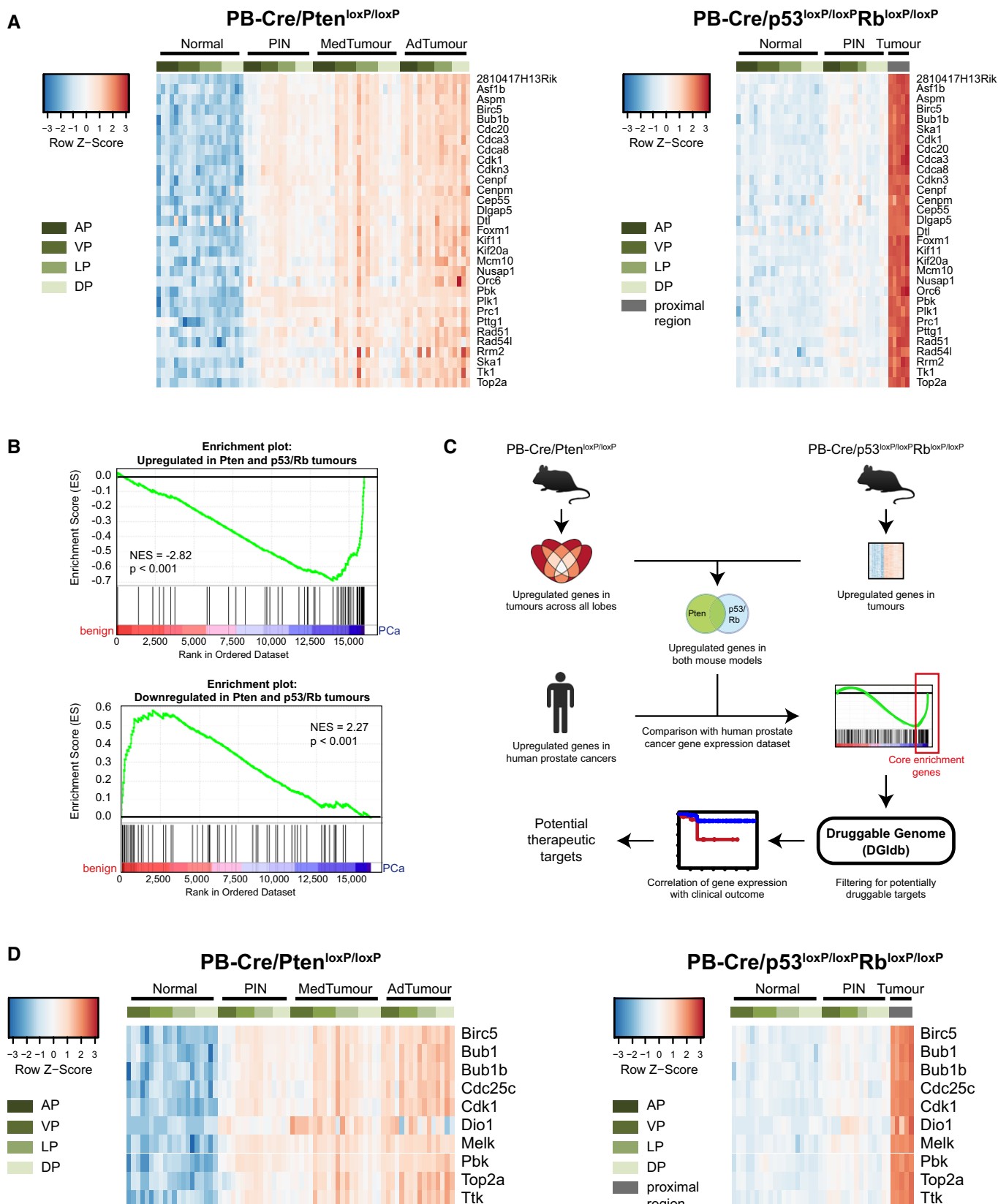

**Figure 3.**

Figure 3.   Identification of potential therapeutic targets in prostate cancer using cross-species analysis.

A   Expression patterns of the Cuzick signature mouse homologous genes in prostate tumours arising in the PB-Cre/Pten$^{loxP/loxP}$ and PB-Cre/p53$^{loxP/loxP}$Rb$^{loxP/loxP}$ models.
B   Gene set enrichment analyses (GSEAs) comparing genes upregulated or downregulated in both mouse models to the Grasso human prostate cancer dataset. NES: normalised enrichment score; PCa: prostate cancer.
C   Strategy to identify potential therapeutic targets in prostate cancer. Genes that are upregulated in cancers of both mouse models ("Pten model (all lobes) and p53/Rb model" in Fig EV1A) are identified, and this consensus signature is then compared to the Grasso human prostate cancer expression dataset using GSEA. Genes forming the core enrichment in this analysis are filtered for druggability using The Drug Gene Interaction Database (DGIdb). The association of the expression of the resulting candidate genes with poor outcome is then used to refine the list.
D   Potential therapeutic targets identified using the strategy outlined in (C). The expression levels of the candidate genes in prostate tumours of the PB-Cre/Pten$^{loxP/loxP}$ and PB-Cre/p53$^{loxP/loxP}$Rb$^{loxP/loxP}$ models are depicted as heatmaps.

Source data are available online for this figure.

MELK-positive non-neoplastic cases displayed exclusively nuclear staining; in contrast, MELK staining in PIN and tumour samples was either exclusively cytoplasmic or both cytoplasmic and nuclear, but rarely exclusively nuclear. Interestingly, a similar pattern was observed when comparing tumours with different degrees of aggressiveness. Cytoplasmic, but not nuclear, expression of MELK was associated with increased Gleason score and higher tumour stage (Fig 4E). Taken together, these results confirmed that MELK protein was indeed overexpressed in prostate cancer, and suggested that cytoplasmic MELK was associated with tumour progression.

## Abrogation of MELK activity represses genes associated with tumour progression and reduces proliferation and viability of prostate cancer cells

We then set out to investigate the effects of MELK abrogation on prostate cancer cells *in vitro*. As MELK is most highly expressed in aggressive prostate tumours, we chose the metastatic, castration-resistant C4-2b cell line as our main model system. A potent small-molecule inhibitor of MELK, OTS167, has been developed (Chung *et al*, 2012) and in our hands inhibited phosphorylation of the known MELK substrate ACC (Beullens *et al*, 2005) at nanomolar concentrations (Fig EV4A), indicating successful inhibition of MELK under these experimental conditions. Treatment with OTS167 also reduced MELK protein levels, which has been previously observed and is attributed to decreased MELK stability due to inhibition of autophosphorylation (Lizcano *et al*, 2004; Badouel *et al*, 2010; Chung *et al*, 2016). We performed RNA sequencing following either knock-down of MELK with two different siRNAs (Fig EV4B) or treatment with OTS167. It is well known that both siRNAs and small-molecule inhibitors can have off-target effects, and indeed, a recent publication indicated that OTS167 can inhibit other kinases (although to a lesser extent than MELK) (Ji *et al*, 2016). Nevertheless, there was considerable overlap between genes that were dysregulated following MELK knock-down and following OTS167 treatment (Fig EV4C). In order to exclude any genes that might potentially be dysregulated due to off-target effects of one of the siRNAs or the inhibitor, we selected genes that were consistently differentially expressed following silencing of MELK with both siRNAs and following treatment with OTS167 at a minimum of one time point (Fig EV4B). We found that genes that were repressed by abrogation of MELK activity (i.e. positively regulated by MELK) were predominantly upregulated in prostate cancer compared to benign tissue and in metastatic compared to primary prostate cancers, confirming our hypothesis that MELK plays a role in cancer

progression (Fig 5A). Furthermore, MELK has previously been shown to be positively regulated by Hes6 (Fig EV4D) and is part of a HES6-associated signature that predicts poor outcome in prostate cancer (Ramos-Montoya *et al*, 2014). About 43% of the genes in this signature were also part of our putative MELK-upregulated gene set (Fig 5B). Pathway analysis revealed that silencing of MELK or treatment with OTS167 primarily resulted in changes in the expression of genes associated with cell cycle regulation (Fig 5C), which was consistent with previous studies suggesting that MELK plays a role in the regulation of mitosis (Davezac *et al*, 2002; Badouel *et al*, 2006). Additional, potentially cancer-relevant pathways that were affected by both MELK knock-down and OTS167 treatment included apoptosis, cytoskeletal rearrangement and DNA damage repair pathways.

Both silencing of MELK using siRNAs and treatment with OTS167 strongly reduced viable cell numbers of C4-2b cells (Fig 5D and E) and its castration-sensitive parental cell line LNCaP (Fig EV4E and F). In a panel of five prostate cancer cell lines (LNCaP, C4-2, C4-2b, PC-3 and DU145) and one non-transformed prostate cell line (PNT1a), OTS167 suppressed cell viability with IC50 values between 7.9 and 53.7 nM (Fig EV4G). This was similar to results reported in other MELK-positive cancer cell line models (Chung *et al*, 2012; Kato *et al*, 2016). Notably, there was a statistically significant correlation between MELK expression and sensitivity to OTS167, supporting the notion that the effects of OTS167 on prostate cancer cell viability are at least partially mediated through MELK (Fig 5F). OTS167 also dramatically reduced the clonogenic ability of C4-2b cells (Fig 5G).

Taken together, these results strongly suggest that MELK promotes the survival of prostate cancer cells and the expression of genes associated with tumour progression.

## Treatment with OTS167 suppresses prostate cancer growth *in vivo*

Having demonstrated that silencing of MELK and treatment with OTS167 greatly reduced prostate cancer cell proliferation and viability *in vitro*, we aimed to evaluate whether targeting this kinase might be a promising strategy to reduce tumour growth *in vivo*. Treatment of mice bearing subcutaneous C4-2b xenografts with OTS167 resulted in a strong reduction in tumour growth as demonstrated by bioluminescence imaging (Fig 6A), measurement of tumour volume (Fig 6B) and final tumour weights (Fig 6C). Staining for cleaved caspase-3 (CC3) revealed that treatment with OTS167 induced apoptosis in these tumours (Fig 6D). To further validate the

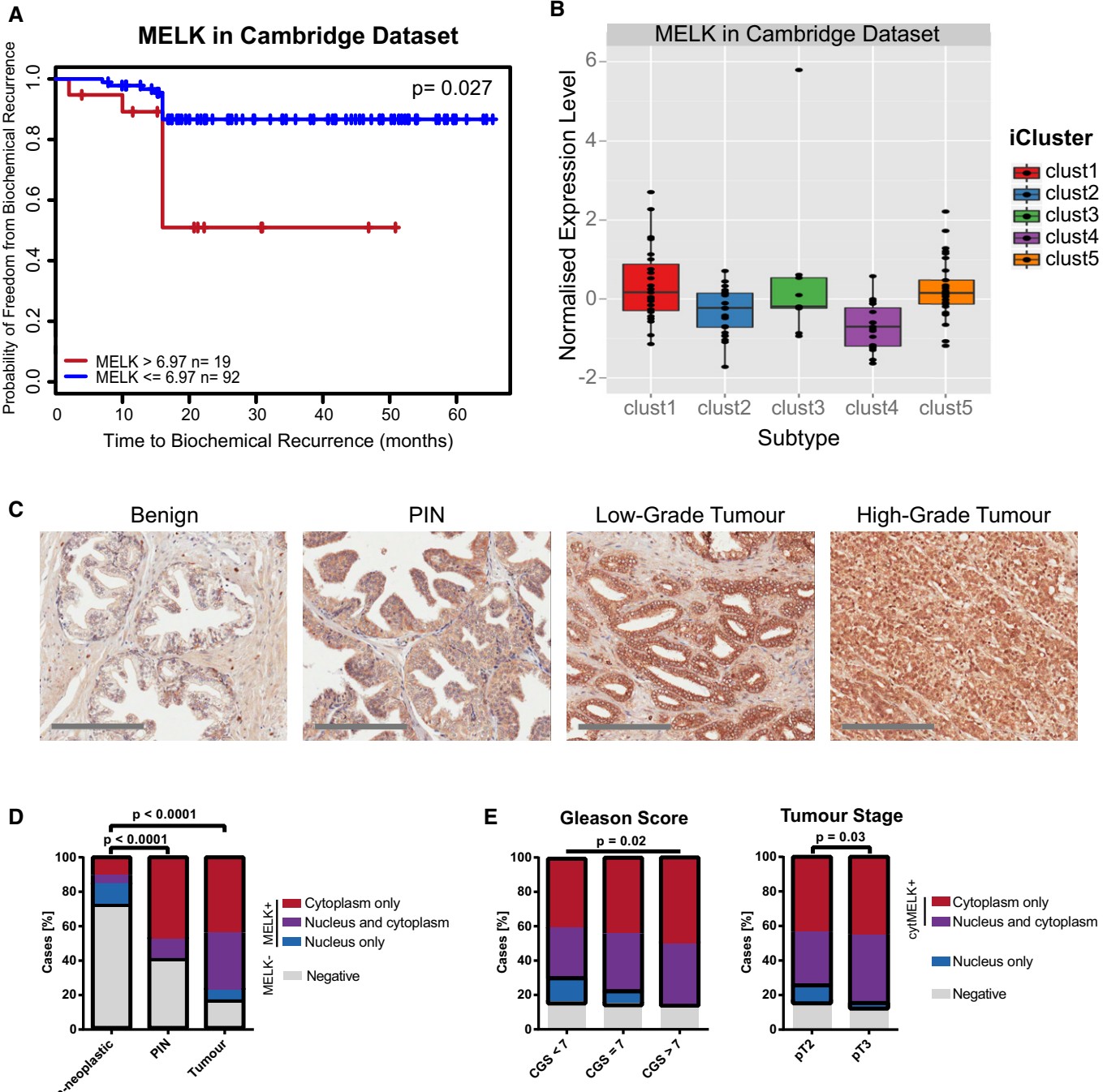

**Figure 4.  MELK is overexpressed in human prostate cancer and associated with an aggressive phenotype.**

A    Kaplan–Meier plot showing that the high expression of MELK is associated with shorter time to relapse in the Cambridge dataset (Ross-Adams *et al*, 2015). Cut-offs for MELK expression levels were determined by recursive partitioning. Statistical significance was assessed by Log-rank test.

B    Expression of MELK in the five prostate cancer iClusters identified in the Cambridge dataset (Ross-Adams *et al*, 2015). Horizontal line indicates median; box limits correspond to 75th and 25th percentiles; whiskers correspond to 95th and 5th percentile.

C    Representative tissue microarray images of immunohistochemical staining for MELK in benign prostate (*n* = 148), PIN (*n* = 38), low-grade and high-grade prostate cancer (*n* = 323). Scale bars correspond to 200 μm.

D    Quantification of immunohistochemical staining for MELK on the tissue microarray. Chi-square test was performed comparing samples with any positive staining for MELK (regardless of localisation) to samples negative for MELK; black outlines indicate groups.

E    Cytoplasmic MELK expression is associated with increased Gleason score and higher tumour stage. Samples with cytoplasmic MELK were tested against samples with nuclear MELK only using chi-square test for Gleason score and Fisher's exact test for tumour stage. CGS: combined Gleason score.

Source data are available online for this figure.

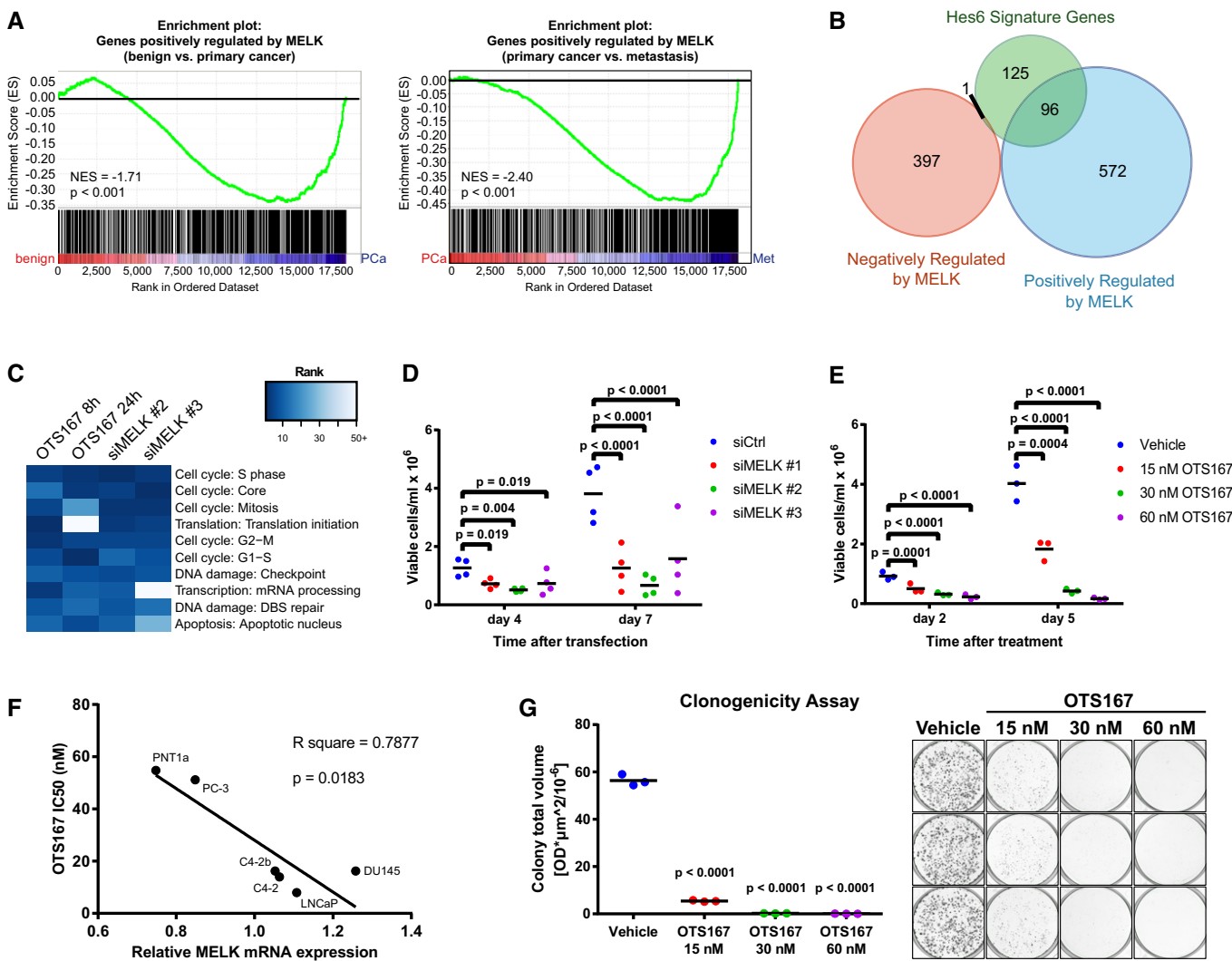

**Figure 5. Abrogation of MELK activity downregulates tumour-relevant genes and suppresses cancer-associated phenotypes.**

A   GSEA comparing genes repressed by both silencing and inhibition of MELK to the Grasso human prostate cancer dataset. C4-2b cells were transfected with siRNAs directed against MELK for 72 h, or treated with 30 nM OTS167 for 8 and 24 h, and subjected to RNA sequencing ($n = 4$). Genes that were significantly downregulated ($P_{adj} < 0.05$) by both siRNAs and by OTS167 at at least one time point were considered as positively regulated by MELK. NES: normalised enrichment score; PCa: prostate cancer; Met: metastases.

B   Overlap between MELK-regulated genes and Hes6 signature (Ramos-Montoya *et al*, 2014).

C   Altered process networks following MELK abrogation. Differentially expressed genes for each treatment condition were identified ($P_{adj} < 0.05$), and the 50 most enriched process networks in each condition were computed using Metacore enrichment analysis. Enriched process networks were ranked according to their *P*-value, and the ranks of the 10 most enriched process networks across all four conditions were visualised as a heatmap.

D   Effect of silencing of MELK on proliferation of prostate cancer cells. C4-2b cells were transfected with siRNAs directed against MELK or a non-targeting control, and viable cells were counted after 4 and 7 days. $n = 4$.

E   Effect of OTS167 on proliferation of prostate cancer cells. C4-2b cells were treated with vehicle or OTS167 at varying concentrations, and viable cells were counted after 2 and 5 days. $n = 3$.

F   Correlation between MELK expression and sensitivity to OTS167. MELK mRNA levels in prostate cell lines were determined by qRT–PCR, $n = 4$ for all cell lines, except C4-2b ($n = 5$). The IC50 for OTS167 in each cell line was determined (see Fig EV4F). The correlation between MELK expression and OTS167 IC50 was assessed by Pearson correlation coefficient.

G   Effect of OTS167 on clonogenic ability of prostate cancer cells. C4-2b cells were seeded at low confluence and grown in presence of vehicle or varying concentrations of OTS167 for 9 days. Colonies were stained with crystal violet, and total colony volume was quantified. $n = 3$.

Data information: Statistical significance was assessed by randomised blocks ANOVA (significance threshold of 0.05) followed by Holm–Sidak's multiple comparisons test in panels (D, E and G).

Source data are available online for this figure.

anti-apoptotic role of MELK in prostate cancer, we treated C4-2b cells with OTS167 *in vitro* and assessed the rate of apoptosis using Annexin V staining. Consistent with our findings *in vivo*, we observed an increase in the fraction of apoptotic cells (Fig 6E). The same effect was observed following silencing of MELK using siRNAs, supporting the interpretation that this effect of the drug

could be mediated through MELK (Fig 6F). In conclusion, treatment with OTS167 induced apoptosis in prostate cancers *in vivo* and abrogated tumour growth.

### Treatment with OTS167 reduces phosphorylation of stathmin and interferes with mitotic spindle formation

To better understand both MELK-dependent and MELK-independent effects of OTS167 that might contribute to its anti-tumour effects *in vitro* and *in vivo*, we performed an antibody array interrogating 674 phosphorylation sites on 400 proteins, using lysates from OTS167-treated C4-2b cells.

We identified 147 phosphorylation sites that passed our quality control and exhibited log2-fold changes of > 0.5 (Table EV2; Fig 7A). Many cancer-relevant signalling pathways were affected by OTS167; one example was the p90RSK pathway, with several activating sites of p90RSK and phosphorylation sites on three of its substrates, including an inhibitory phosphorylation site on the

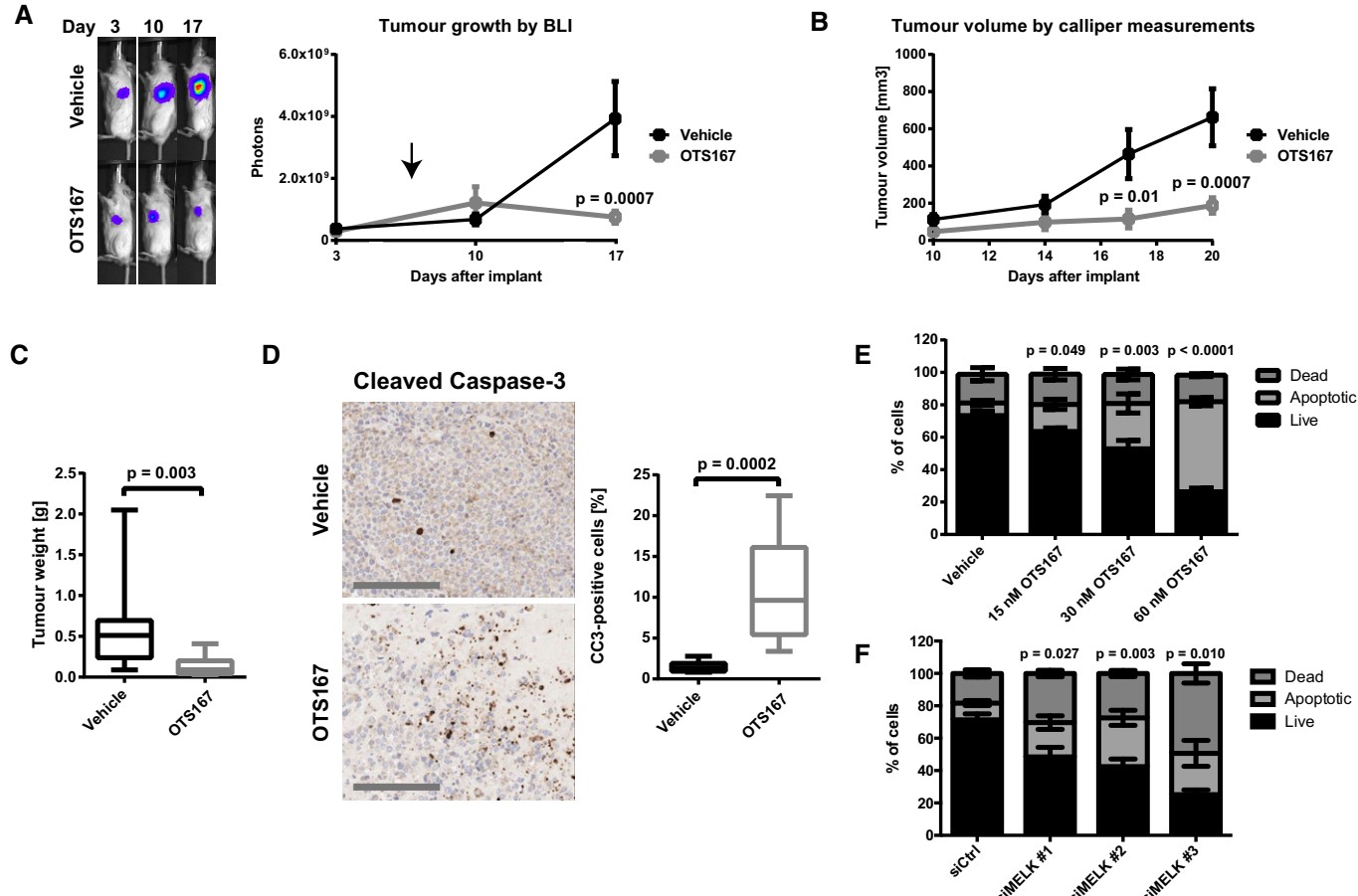

**Figure 6.  OTS167 suppresses prostate cancer growth *in vivo*.**

A   OTS167 reduces growth of prostate cancer xenografts. Luciferase-expressing C4-2b xenograft tumours were established in NOD scid gamma mice for 7 days. Animals were subsequently dosed with 10 mg/kg OTS167 i.p. daily. Bioluminescence was measured once per week. *n* = 10. Statistical significance was assessed using the Holm–Sidak method (α = 5%). Arrow indicates start of dosing with vehicle or OTS167.

B   The growth of xenografts in (A) was followed by calliper measurements twice per week from day 10, when tumours first became palpable. *n* = 10. Statistical significance was assessed using the Holm–Sidak method (α = 5%).

C   Xenograft tumours in (A) were weighed after sacrifice. *n* = 10 for vehicle group, *n* = 8 for OTS167 group. Statistical significance was assessed by Mann–Whitney test. Horizontal line indicates median; box limits correspond to 75th and 25th percentiles; whiskers indicate minimum and maximum.

D   Effect of OTS167 on apoptosis of xenograft tumours. Xenograft tumours were stained for cleaved caspase-3 (CC3) as a read-out for apoptosis induction, and CC3-positive cells were quantified. *n* = 10 for vehicle group; *n* = 8 for OTS167 group. Statistical significance was assessed by Mann–Whitney test. Scale bars correspond to 200 μm. Horizontal line indicates median; box limits correspond to 75th and 25th percentiles; whiskers indicate minimum and maximum.

E   Effect of OTS167 on apoptosis of prostate cancer cells. C4-2b cells were treated with vehicle or OTS167 at varying concentrations for 48 h, and apoptotic, live and dead cells were quantified by Annexin V/propidium iodine staining followed by flow cytometry analysis. Statistical significance of the differences between the proportions of apoptotic cells was tested by randomised blocks ANOVA (significance threshold of 0.05) followed by Holm–Sidak's multiple comparisons test. *n* = 3.

F   Effect of MELK siRNA on apoptosis of prostate cancer cells. C4-2b cells were transfected with siRNAs directed against MELK for 4 days, and apoptotic, live and dead cells were quantified by Annexin V/propidium iodine staining followed by flow cytometry analysis. Statistical significance of the differences between the proportions of apoptotic cells was tested by randomised blocks ANOVA (significance threshold of 0.05) followed by Holm–Sidak's multiple comparisons test. *n* = 4.

Source data are available online for this figure.

pro-apoptotic protein BAD, showing clear reductions (Fig EV5A). This was also confirmed by Western blot (Fig 7B).

The two most significantly reduced phosphorylation events on the antibody array were Ser16 and Ser38 of the microtubule-destabilising protein stathmin (Fig 7A). Stathmin is inhibited by

phosphorylation upon entry into mitosis, enabling the mitotic spindle to form, and subsequently dephosphorylated to allow depolymerisation of the spindle and exit from mitosis (Rubin & Atweh, 2004). Studies using stathmin mutants that could not be phosphorylated have shown that failure to deactivate stathmin upon

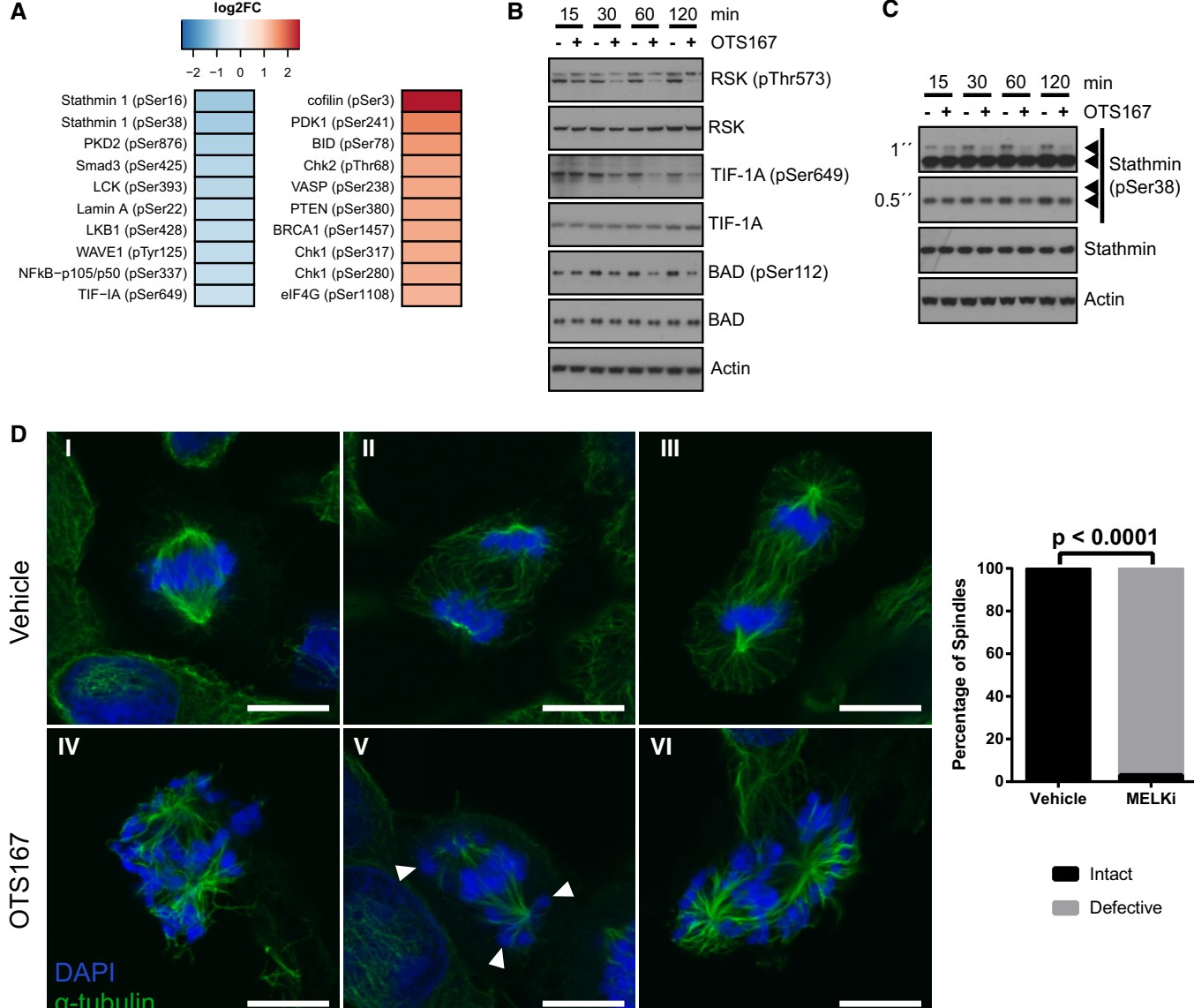

**Figure 7. Inhibition of MELK reduces phosphorylation of stathmin and interferes with mitotic spindle formation.**

A   Identification of phosphorylation sites affected by OTS167. C4-2b cells were treated with vehicle or 30 nM OTS167 for 2 h, and phosphoproteins were analysed using a Phospho Explorer Antibody Array. Signals for each phosphorylation site were normalised to its corresponding total protein. Top upregulated and downregulated phosphorylation sites are shown. $n = 1$.

B   Validation of effects of OTS167 on phosphorylation of p90RSK and its targets. C4-2b cells were treated with vehicle or 30 nM OTS167 for the indicated times, and levels of total and phosphorylated proteins were determined by Western blot analysis. β-Actin was used as a loading control.

C   Treatment with OTS167 reduces phosphorylation of stathmin at Ser-38. C4-2b cells were treated with vehicle or 30 nM OTS167 for the indicated times, and levels of total and phosphorylated stathmin were determined by Western blot analysis. β-Actin was used as a loading control.

D   Treatment with OTS167 results in formation of abnormal mitotic spindles. C4-2b cells were treated with vehicle or 30 nM OTS167 for 24 h. Mitotic spindles and DNA were visualised by immunofluorescent staining for α-tubulin, and by staining with DAPI, respectively. Scale bars correspond to 10 μm. For vehicle-treated cells, examples of normal metaphase (I), anaphase (II) and telophase (III) are shown. For OTS167-treated cells, normal mitotic phases could not be observed. Note mis-attached chromosomes (arrows). Intact and defective spindles were quantified by counting 100 mitotic cells per experimental condition. Significance was assessed using the chi-square test.

Source data are available online for this figure.

mitotic entry results in cell cycle arrest and the formation of abnormal spindles (Segerman *et al*, 2003). We thus hypothesised that OTS167 might interfere with mitotic spindle formation.

To address this hypothesis, we validated the decrease in stathmin phosphorylation induced by OTS167 by Western blot (Fig 7C). We then tested whether OTS167 would induce a similar phenotype of abnormal mitotic spindles as had been described for constitutively active stathmin mutants. Indeed, we observed that the majority of OTS167-treated cells failed to form an ordered metaphase plane and instead exhibited abnormal mitotic spindles showing features characteristic of excess microtubule catastrophe promotion (Segerman *et al*, 2003; Holmfeldt *et al*, 2004), namely disorganised chromosomes, decreased amount of kinetochore microtubules and starlike asters with short, dense microtubules (Fig 7D).

Notably, the reduction in phospho-stathmin following OTS167 treatment was also observed in xenograft tissue, confirming that this effect also occurs in an *in vivo* setting. Due to the small size and extensive cell death of OTS167-treated xenograft tumours, we were only able to extract sufficient amounts of protein from four tumours of this treatment arm. Three out of these four samples showed reduced levels of stathmin phosphorylation compared to samples from vehicle-treated animals (Fig EV5B). Notably, the one sample that did not exhibit decreased phospho-stathmin levels was obtained from an animal whose tumour responded relatively poorly to OTS167 (Fig EV5C).

Interestingly, the known cell cycle-dependent pattern of stathmin phosphorylation correlates well with the previously reported increase in MELK activity during mitosis (Blot *et al*, 2002). Inhibition of stathmin phosphorylation might thus be a MELK-dependent effect of OTS167. Consistent with this notion, silencing of MELK resulted in a similar decrease in stathmin phosphorylation as observed following OTS167 treatment (Fig EV5D).

In conclusion, our data suggest that one of the mechanisms through which OTS167 induces cell death may involve decreased phosphorylation of stathmin, resulting in defective mitotic spindles, and that this may be mediated by MELK.

## Discussion

GEMM of cancer represent powerful tools for understanding tumour biology. The data described in this study represent the most in-depth molecular characterisation of the PB-Cre/Pten[loxP/loxP] model and PB-Cre/p53[loxP/loxP]Rb[loxP/loxP] models of prostate cancer to date. They provide insight into the different stages of tumour progression in these models as well as the similarities and differences of tumours arising in each of the four prostatic lobes.

The main aim of this study was to use the acquired data from GEMM to prioritise potential therapeutic targets for functional validation. Several lines of evidence suggest that this cross-species approach resulted in the identification of genes that have the potential to be therapeutic targets in PCa. Firstly, among the 10 identified candidates, there are a number of genes that are known to play important roles in cancer biology, such as key cell cycle regulators (e.g. Bub1, Bub1b, Cdc25c, Cdk1), and BIRC5 and Top2a. Secondly, a statistically significant association between high expression of the genes and decreased relapse-free survival was found for 6 out of 10 genes. Finally, the protein kinase MELK was chosen for functional

validation to provide proof of principle that potential therapeutic targets were identified.

A number of publications support the notion that MELK is a key regulator of progression and potential therapeutic target in multiple cancer types (Gray *et al*, 2005; Alachkar *et al*, 2014; Wang *et al*, 2014).

We found that MELK is overexpressed in human prostate tumours on the protein level, consistent with a previous study (Kuner *et al*, 2013). Interestingly, we observed that only cytoplasmic, but not nuclear, MELK was associated with parameters of aggressiveness. The underlying mechanisms as well as the functional consequences of this change in MELK localisation will require further investigation. However, one previous study in glioblastoma has already hinted at the possibility of differential functions for nuclear and cytoplasmic MELK (Gu *et al*, 2013).

Our data suggest that inhibition of MELK has the potential to be an effective therapeutic strategy in prostate cancer. We found that treatment with OTS167 reduced the viability of prostate cancer cells at nanomolar concentrations, abrogated growth of xenograft tumours and induced apoptosis of tumour cells both *in vitro* and *in vivo*. The MELK inhibitor used in this study, OTS167, is currently being tested in oncology clinical trials and has been shown to be effective in xenograft models of several cancer types (Chung *et al*, 2012; Wang *et al*, 2014). It has recently been suggested that the anti-proliferative effects of OTS167 on breast cancer cells are mediated through off-target effects, as MELK-knockout cell lines remained sensitive to the inhibitor (Huang *et al*, 2017; Lin *et al*, 2017); however, no such findings have been reported in other cancer types to date. While we cannot fully exclude that off-target effects of OTS167 may have contributed to some of the phenotypes observed in our study, we obtained highly consistent results with both OTS167 and two separate siRNAs directed against MELK, most notably with regard to inhibition of proliferation, induction of apoptosis and reduced phosphorylation of stathmin. We also limited our analysis of the effects of MELK on gene expression to genes that were consistently affected by both OTS167 and two independent siRNAs directed against MELK, which is expected to result in a strong enrichment for *bona fide* MELK-regulated genes.

Other than MELK, only a small number of kinases that are also inhibited by OTS167 have been described so far; interestingly, one of them is the mitotic checkpoint kinase BUB1 (Ji *et al*, 2016). BUB1 was among the 10 potential therapeutic targets that were identified by our cross-species approach (Fig 3D), and its association with poor outcome across multiple independent cohorts was almost as compelling as the one observed for MELK (Fig EV2B). Based on our cross-species data, we would thus predict a compound such as OTS167, which is able to inhibit both MELK and BUB1, to be more effective in targeting prostate cancer cells than single inhibitors of either kinase. A similar approach has already been proposed in kidney cancer cells, where combined targeting of MELK and another kinase, TOPK, was shown to have stronger growth-suppressive effects than targeting either kinase on its own (Kato *et al*, 2016).

We also identified several downstream targets affected by OTS167 in prostate cancer cells, including the pro-apoptotic protein BAD and the microtubule-associated protein stathmin. Interestingly, a previous study found that breast cancer cells treated with OTS167 failed to complete mitosis and subsequently underwent apoptosis (Wang *et al*, 2014). It is possible that loss of stathmin

phosphorylation and the resulting disruption of the mitotic spindle are the underlying mechanism behind this phenotype.

Further studies will be required in order to dissect which of the downstream effects of OTS167 identified in this study are mediated through MELK and which involve other kinase targets. The development of novel, potentially more specific inhibitors of MELK would be an important step in facilitating this research. Efforts in this direction are already underway, and several other molecules have been reported to inhibit MELK; however, so far these are significantly less potent than OTS167 (Huang *et al*, 2017).

Considering that silencing of MELK also decreased stathmin phosphorylation, it would be interesting to confirm whether this is one of the downstream effects of OTS167 that are mediated through MELK. While MELK had previously been shown to play a role in mitosis (Davezac *et al*, 2002; Wang *et al*, 2014), it has not yet been reported to regulate mitotic spindle formation. If confirmed, this would thus constitute a novel function of this kinase. As a next step, it would be worthwhile to test whether stathmin is a direct substrate of MELK. As the consensus phosphorylation motif for MELK is so far unknown, we were unable to determine whether stathmin has a potential MELK phosphorylation site.

Recently, Mu *et al* (2017) and Ku *et al* (2017) showed that prostate tumour progression to anti-androgen resistance involves a mechanism named cellular lineage plasticity, in which the reprogramming transcription factors SOX2 and EZH2 play a role in inducing stemness, neuroendocrine phenotype and canonical androgen receptor signalling independence. Interestingly, MELK activity together with FOXM1 has been shown to positively regulate both SOX2 and EZH2 in other tumours, such as glioma and medulloblastoma (Ganguly *et al*, 2014; Kim *et al*, 2015; Liu *et al*, 2017). Similar complex signalling may occur in advanced stages of PCa, opening new strategies for castration-resistant prostate cancer therapies in the context of resistance to new anti-androgens (enzalutamide) and androgen synthesis inhibitors (abiraterone) (Narayanan *et al*, 2015).

In conclusion, our data demonstrate that cross-species integration of genomic data from cancer GEMM and human cancer patients is a powerful strategy to identify potential therapeutic targets. The protein kinase MELK constitutes an example of a potential therapeutic target that has been predicted using this method and validated using a combination of genomic analyses, survival data and *in vitro* and *in vivo* studies.

## Materials and Methods

### Mice

FVB.129-*Trp53*$^{tm1Brn}$(01XC2) and FVB.129-*Rb1*$^{tm2Brn}$ (01XC1) mice were kindly obtained from the NCI Mouse Models of Human Cancers Consortium (MMHCC) on behalf of Anton Berns (Netherlands Cancer Institute). B6.D2-Tg(Pbsn-cre)4Prb (01XF5) mice were kindly obtained from MMHCC on behalf of Pradip Roy-Burman (University of Southern California). C.129S4-*Pten*$^{tm1Hwu}$/J mice were obtained from The Jackson Laboratory (Bar Harbor, ME). All strains were backcrossed N11 to an FVB/N genetic background. For tumour implantation, immunocompromised (NOD/scid/IL2R gamma null or NSG) male mice (Charles River, Wilmington, MA) were used. Mice were maintained in the Cancer Research UK Cambridge Institute

### The paper explained

**Problem**

Genome-wide expression datasets from human prostate tumours readily allow the identification of genes whose expression levels are altered during cancer development and progression. However, translating this information into clinically useful therapeutic targets poses a twofold challenge: firstly, many genes may be aberrantly expressed as a consequence of cancer development without directly contributing to it, and secondly, not all functionally important genes will represent actionable targets for current drug development approaches.

**Results**

We used genetically modified mouse models of prostate cancer to identify genes that are aberrantly expressed in both human and murine prostate cancers. Through a series of filtering steps, we identified a list of potential therapeutic targets for prostate cancer treatment. We validated one of these potential therapeutic targets, the protein kinase MELK, by showing that silencing of this gene inhibits prostate cancer cell proliferation and induces cell death *in vitro*. Treatment with a compound that inhibits MELK strongly reduced prostate cancer growth *in vivo*.

**Impact**

Our results show that cross-species comparisons of mouse and human prostate cancer gene expression data can identify potential therapeutic targets. Inhibition of one of these potential targets, the kinase MELK, might be a promising strategy for treatment of prostate cancer.

Animal Facility. All experiments were performed in accordance with national guidelines and regulations and with the approval of the animal care and use committee at the institution under UK Home Office project licence 80/2435.

### *In vivo* studies and mouse prostate samples for RNA sequencing

For xenograft studies, two million luciferase-expressing C4-2b cells were injected subcutaneously in the flank of male NSG mice in a 1:1 mix of PBS and phenol red-free HC matrigel (BD, Franklin Lakes, NJ). Tumours were established for 1 week. Following that, mice were dosed daily with 10 mg/kg OTS167 in PBS + 10% DMSO intraperitoneally. Grafts were monitored biweekly by calliper measurement and weekly by imaging after intraperitoneal injection of D-luciferin 150 mg/kg (Caliper Life Sciences, Hopkinton, MA). Luminescent measurements were analysed using Xenogen Imaging Analysis software "Living Image® 3.0" (Caliper Life Sciences) and plotted as photons/second for graphic analysis of growth kinetics. Mice were culled at completion of experiment or when tumours reached 10% of body weight. To select prostate tumour samples, prostate lesions were histopathologically classified according to recommendations from the Mouse Models of Human Cancer Consortium Prostate Pathology Committee (Shappell *et al*, 2004; Ittmann *et al*, 2013). Human genome version hg19 or the mouse genome version mm10 was used to align sequencing reads.

### Gene set enrichment and survival analyses

Gene set enrichment analysis (GSEA) was performed using the GSEAPreranked tool within the GSEA software (Broad Institute,

http://www.broadinstitute.org/gsea/index.jsp; Mootha *et al*, 2003; Subramanian *et al*, 2005), and process networks enriched among gene sets of interest were identified with the one-click process networks enrichment analysis tool within MetaCore™ (Thomson Reuters, Cambridge, UK). Survival analysis was carried out using a Galaxy-based (https://galaxyproject.org/) tool developed by the Cancer Research UK Cambridge Institute Bioinformatics Core Facility.

### Cell lines and cell-based assays

LNCaP, PC-3 and DU-145 cells were obtained from ATCC (Manassas, VA, USA), C4-2 and C4-2b from MD Anderson Cancer Center (Houston, TX, USA). OTS167 was sourced from Haoyuan Chemexpress (Shanghai, China) and dissolved in DMSO. siRNAs against MELK (MELK siRNA #1: Qiagen SI02224558; MELK siRNA #2: Qiagen SI02224565; MELK siRNA #3: Dharmacon J-004029-06) were transfected (final 20 nM) using Lipofectamine RNAiMAX (Invitrogen). AllStars Negative Control siRNA (Qiagen SI03650318) was used as a non-targeting control. Cell viability and proliferation were assessed using Vi-CELL and MTS assays. Apoptotic cells were quantified based on Annexin V and PI staining using a FACS Calibur (BD Biosciences). Clonogenicity assays were quantified using a GelCount imaging system.

### Antibody arrays, western blot, qRT–PCR, immunohistochemistry and immunofluorescence

Antibody arrays were analysed using Phospho Explorer Antibody Arrays (Full Moon Biosystems, Sunnyvale, CA, USA). Western blots were performed for anti-RRN3 pSer649 (ab138651) and anti-β-actin (ab6276), both from Abcam, Cambridge, UK; anti-stathmin pSer38 (4191), anti-p90RSK pThr573 (9346), anti-Bad pSer112 (9291), anti-RSK1/2/3 (9355), anti-Bad (9292) and anti-stathmin (3352), all from Cell Signalling Technology, Cambridge, UK; anti-α-tubulin (T9026) and anti-MELK (HPA017214), all from Sigma-Aldrich, St Louis, MO, USA; anti-RRN3 (sc-133978), from Santa Cruz Biotechnology, Dallas, TX, USA; and anti-MELK (NBP1-19598), from Novus Biologicals, Littleton, CO, USA. IF was performed for α-tubulin (DM1A, Sigma) and IHC for MELK (NBP1-19598, Novus Biologicals, Littleton, CO, USA) and CC3 (9664, Cell Signalling Technology). qRT–PCRs were performed on an ABI PRISM 7900 HT Sequence Detection System. Relative gene expression was calculated according to the ΔΔCt method; HPRT was used as housekeeping gene.

### Data analysis and graphical representation

Statistical analysis was performed in GraphPad Prism6 unless otherwise indicated. Parametric tests were used in cases where normal distribution could be assumed based on prior similar studies; otherwise, non-parametric tests were used. Details of statistical tests used and number of replicates are given in the respective figure legends. All error bars indicate standard error of mean unless otherwise stated, and $n$ indicates independent biological replicates. No power analysis was done *a priori* of study design, since the effect size in changes was unknown.

Hypergeometric tests were performed in R version 3.1.1 (R Core Team, 2014) using the phyper function in the stats package.

Heatmaps were generated using the gplots and RColorBrewer packages. In case of RNAseq data, normalised counts were used as the input.

### Data availability

Study data are deposited in NCBI Gene Expression Omnibus (https://www.ncbi.nlm.nih.gov/geo/) under accession numbers GSE94570 (MELK-regulated genes) and GSE94574 (mouse prostate lobe gene expression).

### Additional methods

Expanded methodology and details of reagents used for *in vivo* studies, selection and preparation of mouse prostate samples for RNA sequencing, gene set enrichment analysis, survival analysis, tissue microarray, cell lines and culture conditions, transfection, viability, apoptosis, clonogenicity, antibody array, Western blot, immunofluorescence, immunohistochemistry, qRT–PCR, data analysis and graphical representation, antibodies, primers, probes and oligonucleotides are described in Appendix Supplementary Methods.

**Expanded View** for this article is available online.

### Acknowledgements

We thank the staff of the core facilities at the Cancer Research UK Cambridge Institute for their support with this work: Genomics, Bioinformatics, Research Instrumentation, Light Microscopy, Histopathology, Flow Cytometry and the Biological Resources Unit. This work was supported by a Cancer Research UK project grant (DEN). SLF received a grant and fellowship from the Sao Paulo Research Foundation (FAPESP) ref. 2013/08830-2 and 2013/06802-1 and from CNPq ref. 305391/2014-3. We would like to extend our acknowledgements to the University of Cambridge, Cancer Research UK and Hutchinson Whampoa Limited.

### Author contributions

SJ and AR-M conducted most of the experiments and analysis and wrote the manuscript. CS, NP-G, MD, JA, KW, ADL and SLF generated data and performed analyses or both. JSC, LGDF, SLF and DEN supervised the project. SLF and DEN were responsible for overseeing the manuscript. All authors discussed the results and reviewed the manuscript.

### Conflict of interest

The authors declare that they have no conflict of interest.

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
