## [Review Process File · EMBO Molecular Medicine]

Identification of Potential Therapeutic Targets in Prostate Cancer through a Cross-Species Approach

Sarah Jurmeister, Antonio Ramos-Montoya, Chiranjeevi Sandi, Nelma Pértega-Gomes, Karan Wadhwa, Alastair D. Lamb, Mark Dunning, Jan Attig, Jason S. Carroll, Lee G.D. Fryer, Sérgio L. Felisbino and David E. Neal

Review timeline:

Submission date:	15 July 2017
Editorial Decision:	01 September 2017
Revision received:	26 November 2017
Editorial Decision:	20 December 2017
Revision received:	01 January 2018
Accepted:	08 January 2018

Editors: Roberto Buccione and Céline Carret

Transaction Report:

1st Editorial Decision

01 September 2017

Thank you for the submission of your manuscript to EMBO Molecular Medicine. We are very sorry that it has taken so long to get back to you on your manuscript.

In this case we experienced unusual difficulties in securing three willing and appropriate reviewers due in part to the overlap with the holiday season. Further to this, reviewer #1 ultimately failed to deliver his/her report notwithstanding repeated chasers and his/her assurances that s/he would deliver. As a further delay cannot be justified I have decided to proceed based on the two available consistent evaluations.

As you will see, although both Reviewers find the study of interest a number of concerns are expressed, with reviewer 3 being more reserved. I will not go into detail as the comments are clear and sufficiently detailed. I would just like to mention one critical issue. Specifically, reviewer 3 notes that it remains unproven that OTSSP167 is actually acting as a MELK inhibitor in the context of your experimental settings. The reviewer suggests a number of avenues to address this crucial issue, which impinges directly on the translational relevance of this work.

While publication of the paper cannot be considered at this stage, we would be pleased to consider a revised submission, with the understanding that the Reviewers' concerns must be fully addressed including with additional experimental data where appropriate and that acceptance of the manuscript will entail a second round of review.

I look forward to seeing a revised form of your manuscript in due time.

***** Reviewer's comments *****

Referee #2 (Remarks for Author):

The authors Jurmeister et al. have mined RNA-sequencing data generated from two distinct murine models of prostate cancer and cross-referenced the differentially-expressed genes with human datasets to identify novel druggable targets. While the murine models themselves are not novel, this approach has led them to identify a promising and under-studied kinase (MELK) that is required for prostate cancer cell survival and tumor growth. The authors provide convincing evidence that treatment with a MELK inhibitor leads to prostate cancer cell death and significantly slows tumor growth in a xenograft model. Jurmeister et al. also present data to suggest a possible novel mechanism of function for MELK inhibition: to disrupt mitotic spindle formation through the protein stathmin. These claims show significant promise for the use of MELK inhibitors to treat prostate cancers with MELK overexpression. This paper stands out from others in the field because the authors have utilized robust bioinformatics analysis and multiple approaches to demonstrate their findings. The text is written clearly. There are a few comments the authors should address to further improve the manuscript. With these edits, the manuscript should be accepted for publication.

1. The authors should elaborate on their findings in the context of the recent studies published on TP53/RB1 altered tumors.
Ku et al. Rb1 and Trp53 cooperate to suppress prostate cancer lineage plasticity, metastasis, and antiandrogen resistance. 2017. Science.
Mu et al. SOX2 promotes lineage plasticity and antiandrogen resistance in TP53- and RB1-deficient prostate cancer. 2017. Science.
2. It is not clear why the authors chose to focus on the RNAseq from the Pten^{-/-} mice and not the p53^{-/-};Rb^{-/-} mice in Figure 2. A better justification for this in the text is needed.
 - a. Note: labeling of AdT-specific genes is unclear in Fig. 2E
3. The two distinct mouse models are not clearly labeled in Supplemental Fig. 4, and an image of PIN lesions for the p53^{-/-};Rb^{-/-} model as described in the text is lacking.
4. Could the authors comment on the age to X phenotype (PIN, MedTumor, AdTumor) for each of the mouse models? Although these are well-studied models, this information would allow the reader to better place this study in the context of the field.
5. The labeling of Supplemental Fig. 6E is unclear. It is difficult to tell which line represents MELKi 4, and there are unlabeled data points as well.

Referee #3 (Comments on Novelty/Model System for Author):

The authors do a good job of describing the differences between the mouse and human prostate and the two different genetically engineered mouse models they use in their studies.

Referee #3 (Remarks for Author):

The manuscript by Ramos-Montoya, et al., is well written with a sound experimental approach to identify novel therapeutic targets in prostate cancer (PCa). Their approach started with rational comparisons between two transgenic mouse models of PCa (PB-Cre/p53PRb and PB-Cre/PTEN), including differences of tumor grade and location within the mouse prostate where tumors form. The authors also nicely describe anatomical differences between mice & human prostates as a rationale for including tumors from multiple regions of the mouse prostate in each of the model systems for further analysis. Using RNAseq expression profiling and a set of bioinformatics-based filtering steps to reduce the complexity of the differentially expressed genes and to help identify changes that are likely to be therapeutically targetable, the authors found MELK to be significantly associated with more aggressive tumors and poor outcome in human patients. The authors then go on to show, in several human cell line models of PCa, that abrogating MELK expression using siRNA or adding a putative inhibitor of MELK (MELKi) activity inhibits cellular proliferation and induces apoptosis and that treating mice harboring cell line xenografts with the MELKi slowed tumor growth and increased apoptosis within the tumors.

The work identifying MELK as a potential therapeutic target in PCa is very thorough and the results and interpretations are sound. However, the studies conducted to validate MELK as a therapeutic target in

PCa rely heavily on the activity and specificity of the chosen MELKi (OTSSP167). Although many of the results obtained using the MELKi were similar to the results obtained with MELK-targeting siRNA, this correlation is insufficient support that the putative MELKi is, in fact, inhibiting MELK activity. Moreover, the authors do not describe the source of the MELKi nor do they provide any supporting evidence of its specificity toward MELK. In fact, the authors themselves acknowledge the recent report that seems to invalidate the presumed dependency on MELK of several cell line models (<https://elifesciences.org/articles/24179>) and suggests the antiproliferative activity of OTSSP167 has substantial effects on targets other than MELK. Thus, the inference that MELK regulates mitotic spindle formation is not sufficiently proven since OTSSP167-treated cells were used to draw this conclusion.

Due to the uncertain specificity of OTSSP167 toward MELK, several questions should be considered in order to bolster the authors' findings of the importance of MELK in PCa aggressiveness and its utility as a therapeutic target. 1) Does the converse experiment (overexpression of MELK in a low-expressing cell line vs abrogation of expression in a high-expressing cell line) result in the upregulation of the same genes and increase aggressiveness? 2) Does the expression of genes identified as MELK-regulated get modulated similarly in response to OTSSP167 treatment of cells that do not express MELK? 3) Does decreasing cellular proliferation by another means, e.g. androgen deprivation of androgen-dependent CaP cells, result in similar expression modulation of the same genes? 4) Are there any CaP cell lines that do not respond to MELKi or siMELK?

Minor points/suggestions:

1. A graphical overview of the approach used to identify MELK as a target for PCa would be useful.
2. Experimental methods section (and other places in the main text) should refer to Supplementary Text for relevant information.
3. White areas in legends of Fig 4D & E are not represented in graph, challenging rapid interpretation.

1st Revision - authors' response

26 November 2017

Reviewer's comment:

The authors should elaborate on their findings in the context of the recent studies published on TP53/RB1 altered tumors.

Ku et al. Rb1 and Trp53 cooperate to suppress prostate cancer lineage plasticity, metastasis, and antiandrogen resistance. 2017. Science.

Mu et al. SOX2 promotes lineage plasticity and antiandrogen resistance in TP53- and RB1-deficient prostate cancer. 2017. Science.

Authors' response:

A new paragraph has been included in the discussion section discussing these recent publications and commenting on the possible role that MELK inhibition could play in the context of prostate cancer progression towards antiandrogen therapy resistance due to cellular lineage plasticity, neuroendocrine and stem cell phenotype acquisition.

Reviewer's comment:

It is not clear why the authors chose to focus on the RNAseq from the Pten-/- mice and not the p53-/-;Rb-/- mice in Figure 2. A better justification for this in the text is needed.

Authors' response:

The reason that Figure panels 2C to 2E focus on the *Pten*^{-/-} data is that the analyses conducted would not have been feasible with the data obtained from *p53*^{-/-};*Rb*^{-/-} mice. Only a very small number of differentially expressed genes were identified in PIN lesions from *p53*^{-/-};*Rb*^{-/-} compared to normal prostate lobes (between 25 and 63 genes depending on the lobe, at a significance level of 0.01). We believe that this is because the *p53*^{-/-};*Rb*^{-/-} mice developed low-grade PIN lesions, which did not accumulate many gene expression alterations. The analyses described in Figure panels 2C to 2E, e.g. pathway analysis, would not have been informative with such a small number of differentially expressed genes. Furthermore, we were unable to differentiate similarly distinct stages of tumour progression in the *p53*^{-/-};*Rb*^{-/-} mice as we did in the *Pten*^{-/-} mice (e.g. medium- and advanced-stage tumours). In our hands, the *Pten*^{-/-} model was thus better suited to exploring how gene expression patterns differ between different prostate lobes and stages of prostate cancer progression, and we have now revised the manuscript to explain this.

It is worth noting that the subsequent analyses, including the cross-species analyses aimed at identifying potential therapeutic targets, considered both mouse models to an equal extent, and we found that this was valuable as it improved the overlap between the mouse and human data (Figure EV1B).

Reviewer's comment:

Note: labeling of AdT-specific genes is unclear in Fig. 2E

Authors' response:

We thank the reviewer for bringing this to our attention and have corrected the labelling to be consistent with the other figures.

Reviewer's comment:

The two distinct mouse models are not clearly labeled in Supplemental Fig. 4, and an image of PIN lesions for the p53^{-/-};Rb^{-/-} model as described in the text is lacking.

Authors' response:

We thank the reviewer for bringing this to our attention. Supplemental Figure 4 (now referred to as Figure EV3 in this resubmission) has been revised to include labels indicating the two distinct mouse models. The image showing PIN lesions for the the p53^{-/-};Rb^{-/-} model that was inadvertently omitted in the first version of this figure has also been added.

Reviewer's comment:

Could the authors comment on the age to X phenotype (PIN, MedTumor, AdTumor) for each of the mouse models? Although these are well-studied models, this information would allow the reader to better place this study in the context of the field.

Authors' response:

This information has been added to the beginning of the results section.

Reviewer's comment:

The labeling of Supplemental Fig. 6E is unclear. It is difficult to tell which line represents MELKi 4, and there are unlabeled data points as well.

Authors' response:

This figure (referred to as Figure EV5 in this resubmission) has been edited for clarity.

Response to Reviewer #3

Reviewer's comment:

The work identifying MELK as a potential therapeutic target in PCa is very thorough and the results and interpretations are sound. However, the studies conducted to validate MELK as a therapeutic target in PCa rely heavily on the activity and specificity of the chosen MELKi (OTSSP167). Although many of the results obtained using the MELKi were similar to the results obtained with MELK-targeting siRNA, this correlation is insufficient support that the putative MELKi is, in fact, inhibiting MELK activity. Moreover, the authors do not describe the source of the MELKi nor do they provide any supporting evidence of its specificity toward MELK. In fact, the authors themselves acknowledge the recent report that seems to invalidate the presumed dependency on MELK of several cell line models (<https://elifesciences.org/articles/24179>) and suggests the antiproliferative activity of OTSSP167 has substantial effects on targets other than MELK. Thus, the inference that MELK regulates mitotic spindle formation is not sufficiently proven since OTSSP167-treated cells were used to draw this conclusion. Due to the uncertain specificity of OTSSP167 toward MELK, several questions should be considered in order to bolster the authors' findings of the importance of MELK in PCa aggressiveness and its utility as a therapeutic target. 1) Does the converse experiment (overexpression of MELK in a low-expressing cell line vs abrogation of expression is a high-expressing cell line) result in the upregulation of the same genes and increase aggressiveness? 2) Does the expression of genes identified as MELK-regulated get modulated similarly in response to OTSSP167 treatment of cells that do not express MELK? 3) Does decreasing cellular proliferation by another means, e.g. androgen deprivation of androgen-dependent CaP cells, result in similar expression modulation of the same genes? 4) Are there any CaP cell lines that do not respond to MELKi or siMELK?

Authors' response:

We thank the reviewer for these important comments and suggestions, and we agree that the specificity of the compound used to inhibit MELK is a key consideration, considering that off-target effects are

commonly observed with kinase inhibitors. In the revised manuscript as well as in this response, we are thus presenting additional data and context to support the conclusions drawn in our study.

The MELK inhibitor used in this study, OTS167, was first described by Chung and colleagues (Chung *et al*, 2012). In their study, the ability of OTS167 to inhibit MELK was demonstrated using *in vitro* kinase assays. Chung and colleagues also tested in their study the growth inhibitory effect of OTS167 in several different cancer cell lines and found that cells with low MELK expression are much less sensitive to growth inhibition by OTS167 than cells with high MELK expression.

We have revised Figure EV4 to include data supporting that OTS167 also inhibits MELK at the concentrations and in the experimental system used in our study. Treatment of C4-2b cells with OTS167 reduced the phosphorylation of ACC at Ser79 (revised Figure EV4A), a known MELK substrate, in a dose-dependent manner (Beullens *et al*, 2005). Furthermore, OTS167 treatment also reduced MELK protein levels (revised Figure S5A), which has been previously observed and attributed to decreased MELK stability due to inhibition of autophosphorylation (Lizcano *et al*, 2004; Badouel *et al*, 2010; Chung *et al*, 2016). Taken together, these results support the conclusion that OTS167 does indeed inhibit MELK activity under the experimental conditions used in this study. We have also added the source of OTS167 used in our experiments to the methods section, which was inadvertently omitted in the original version of this manuscript.

We fully agree with the reviewer that studying the effects of MELK overexpression in a low-expressing prostate cancer cell line would be a worthwhile experimental approach, and indeed increased aggressiveness following overexpression of MELK has been previously demonstrated in breast cancer (Wang *et al*, 2014). However, despite our best efforts we were unable to identify a prostate cancer cell line that could serve as a suitable model system.

We initially investigated five prostate cancer cell lines (LNCaP, C4-2, C4-2b, PC-3, DU145) and one nontransformed prostate cell line (PNT1a) that are regularly used in our laboratory. All six of these cell lines exhibited robust expression of MELK, with only relatively minor differences between cell lines (Response Figure 1); MELK expression levels as assessed by qPCR were less than 2-fold higher in the highest-expressing cell line (DU145) than in the lowest-expressing cell line (PNT1a).

Response Figure 1: MELK expression in prostate-derived cell lines. MELK mRNA is robustly expressed across a panel of prostate cancer cell lines. mRNA was isolated from five prostate cancer cell lines and one immortalised non-transformed prostate line (PNT1a), and expression levels of MELK were quantified by qRT-PCR. Expression levels are shown relative to MELK expression levels in a mix of template from all cell lines. n = 4 for all cell lines, except C4-2b (n = 5).

In an effort to identify a more suitable MELK low-expressing cell line, we retrieved data on MELK expression from the Cancer Cell Line Encyclopedia (<https://portals.broadinstitute.org/ccle>). MELK expression data was available for eight prostate cancer cell lines (NCIH660, VCaP, MDAPCA2B, DU145, LNCaP, 22RV1, PC3, PRECLH), five of which were not included in our own cell line panel. A comparison of MELK expression in prostate cancer cell lines with other cell lines for which data is

available in the Cancer Cell Line Encyclopedia illustrates that MELK expression in all eight prostate cancer cells is comparatively high overall and displays relatively little variation between cell lines (Response Figure 2). In contrast to many other cancer types, e.g. breast, stomach and melanoma, there are no clear “outliers” with low MELK expression among prostate cancer cell lines that are likely to be promising models to test the effect of MELK overexpression.

Response Figure 2: MELK expression in cell lines. Data obtained from the Cancer Cell Line Encyclopedia (<https://portals.broadinstitute.org/ccle>).

The reviewer also raises the question of whether there are any prostate cancer cell lines that do not respond to siMELK or treatment with OTSSP167. In our laboratory, we have so far tested the growth-inhibitory effect of siMELK in LNCaP, C4-2, C4-2b and PNT1a cells, and have observed reduction of proliferation and decreased cell viability in all cases (Figure 5D, Figure EV4E, Response Figure 3). Consistent with this, all prostate cell lines tested to date (LNCaP, C4-2, C4-2b, PC-3, DU145, PNT1a) were sensitive to treatment with OTS167. These results are not surprising, considering that all of these cell lines exhibit robust MELK expression as outlined above. Interestingly, despite the relatively modest differences in MELK expression, we did observe a statistically significant correlation between MELK expression levels and sensitivity to OTS167 in the six cell lines, which would be consistent with the interpretation that the growth inhibitory effects of OTS167 may be mediated by MELK. These results have now been incorporated into Figure 5F.

Response Figure 3: Effect of siMELK on growth of C4-2 and PNT1a cells. C4-2 cells (left) or PNT1a cells (right) were transfected with siRNAs directed against MELK or a non-targeting control, and viable cells were counted at the time points indicated. Preliminary data - n = 2 for C4-2, n = 1 for PNT1a, with three technical replicates per biological replicate.

In order to address the reviewer's question whether inhibition of cell proliferation by other means results in expression modulation of the same genes as treatment with OTS167, we used microArray and RNA-seq data of LNCaP and C4-2b cells treated with established growth-inhibitory compounds: the androgen-inhibitors enzalutamide (Wang *et al*, 2016) or bicalutamide (unpublished data from our laboratory) and the AMPK activators AICAR and metformin 24 h (Jurmeister *et al*, 2014). To facilitate cross-comparability, we only used genes covered in all datasets (11,210 genes), averaged microArray data across all probes of the same gene after inter-quartile normalisation, and used a moderated log₂ fold change estimate for RNAseq data. We then used principal component analysis (PCA) to discern systemic differences between treatments. As shown in Response Figure 4, the gene expression profile of cells transfected with siMELK #2 most closely resembled that of cells treated with OTS167 for 24 h, and the gene expression profile of cells transfected with siMELK#3 most closely resembled that of cells treated with OTS167 for 8 h. By contrast, there was greater variance between MELK knock-down or OTS167 treatment and the other growth-inhibitory stimuli. This suggests that silencing of MELK and treatment of OTS167 result in relatively similar gene expression profiles compared to unrelated treatment conditions.

Response Figure 4: Principal component analysis of prostate cancer gene expression following growth inhibition. To facilitate cross-comparability, only genes covered in all conditions (11,210 genes), were selected. microArray data was averaged across all probes of the same gene after inter-quartile normalisation. A moderated log₂ fold change estimate was used for RNAseq data. PCA was used to discern systemic differences between treatments.

The totality of data presented above and in our revised manuscript continues to support the conclusion that the growth-inhibitory effects of OTS167 in prostate cancer cells are at least in part mediated through MELK:

- OTS167 inhibits MELK under the experimental conditions used in the study, as evidenced by reduced phosphorylation of a MELK substrate and decreased MELK protein levels.
- MELK expression positively correlates with sensitivity to OTS167 in a panel of prostate cell lines.
- Treatment with OTS167 and silencing of MELK both result in similar changes in the expression of cancer-relevant genes, and the resulting gene expression profile is distinct from that induced by unrelated growth-inhibitory compounds.
- Growth inhibition and induction of apoptosis are not only observed following treatment with OTS167, but also following siRNA-mediated knock-down of MELK.

Nonetheless, we have revised the discussion section of the manuscript in order to accurately reflect recent literature indicating that, like most kinase inhibitors, OTS167 inhibits more than one kinase (Ji *et al*, 2016), and to discuss the potential implications for our study. To avoid giving the impression of complete specificity for MELK in absence of data to this effect, we have also changed all references to the inhibitor in the text and figures from “MELK” to “OTS167”.

Finally, we acknowledge the reviewer's point that further experiments will be required in order to determine whether the effect of OTS167 on mitotic spindle formation is mediated through MELK or through another target of the inhibitor, and we have now revised the text and figures of our manuscript to reflect this. Nevertheless, we feel that this does not significantly impact the main conclusions of the study, namely that the cross-species approach described in the manuscript is able to identify potential therapeutic targets in prostate cancer, of which MELK serves as one example.

Reviewer's comment:

A graphical overview of the approach used to identify MELK as a target for PCa would be useful.

Authors' response:

We agree with the reviewer and have revised Figure 3C to show a graphical overview of the steps used to derive potential therapeutic target genes for prostate cancer and identify MELK.

Reviewer's comment:

Experimental methods section (and other places in the main text) should refer to Supplementary Text for relevant information.

Authors' response:

We have revised the main text in line with the reviewer's suggestion.

Reviewer's comment:

White areas in legends of Fig 4D & E are not represented in graph, challenging rapid interpretation.

Authors' response:

We have revised the legend of Figure 4D and E and hope that this will aid interpretation of the figure.

References:

- Badouel C, Chartrain I, Blot J & Tassan JP (2010) Maternal embryonic leucine zipper kinase is stabilized in mitosis by phosphorylation and is partially degraded upon mitotic exit. *Exp. Cell Res.* **316**: 2166–2173 Available at: <http://dx.doi.org/10.1016/j.yexcr.2010.04.019>
- Beullens M, Vancauwenbergh S, Morrice N, Derua R, Ceulemans H, Waelkens E & Bollen M (2005) Substrate specificity and activity regulation of protein kinase MELK. *J. Biol. Chem.* **280**: 40003–11 Available at: <http://www.ncbi.nlm.nih.gov/pubmed/16216881> [Accessed December 15, 2014]
- Chung S, Kijima K, Kudo A, Fujisawa Y, Harada Y, Taira A, Takamatsu N, Miyamoto T, Matsuo Y & Nakamura Y (2016) Preclinical evaluation of biomarkers associated with antitumor activity of MELK inhibitor. *Oncotarget* **7**: 18171–18182 Available at: <http://www.ncbi.nlm.nih.gov/pubmed/26918358> [Accessed October 13, 2017]
- Chung S, Suzuki H, Miyamoto T, Takamatsu N, Tatsuguchi A, Ueda K, Kijima K, Nakamura Y & Matsuo Y (2012) Development of an orally-administrative MELK-targeting inhibitor that suppresses the growth of various types of human cancer. *Oncotarget* **3**: 1629–40 Available at: <http://www.pubmedcentral.nih.gov/articlerender.fcgi?artid=3681500&tool=pmcentrez&rendertype=abstract> [Accessed December 15, 2014]
- Ji W, Arnst C, Tipton AR, Bekier ME, Taylor WR, Yen TJ & Liu S-T (2016) OTSSP167 Abrogates Mitotic Checkpoint through Inhibiting Multiple Mitotic Kinases. *PLoS One* **11**: e0153518 Available at: <http://dx.plos.org/10.1371/journal.pone.0153518> [Accessed November 3, 2017]
- Jurmeister S, Ramos-Montoya A, Neal DE & Fryer LGD (2014) Transcriptomic analysis reveals inhibition of androgen receptor activity by AMPK in prostate cancer cells. *Oncotarget* **5**: 3785–99 Available at: <http://www.pubmedcentral.nih.gov/articlerender.fcgi?artid=4116520&tool=pmcentrez&rendertype=abstract>
- Lizcano JM, Göransson O, Toth R, Deak M, Morrice N a, Boudeau J, Hawley S a, Udd L, Mäkelä TP, Hardie DG & Alessi DR (2004) LKB1 is a master kinase that activates 13 kinases of the AMPK subfamily, including MARK/PAR-1. *EMBO J.* **23**: 833–43 Available at: <http://www.pubmedcentral.nih.gov/articlerender.fcgi?artid=381014&tool=pmcentrez&rendertype=abstract> [Accessed December 15, 2014]
- Wang J, Zou JX, Xue X, Cai D, Zhang Y, Duan Z, Xiang Q, Yang JC, Louie MC, Borowsky AD, Gao AC, Evans CP, Lam KS, Xu J, Kung H-J, Evans RM, Xu Y & Chen H-W (2016) ROR- γ drives androgen receptor expression and represents a therapeutic target in castration-resistant prostate cancer. *Nat. Med.* **22**: 488–496 Available at: <http://www.ncbi.nlm.nih.gov/pubmed/27019329> [Accessed November 19, 2017]
- Wang Y, Lee Y, Baitsch L, Huang A & Xiang Y (2014) MELK is an oncogenic kinase essential for mitotic progression in basal-like breast cancer cells. *Elife* Available at: <http://europepmc.org/abstract/MED/24844244> [Accessed December 22, 2014]

2nd Editorial Decision

20 December 2017

Thank you for the submission of your revised manuscript to EMBO Molecular Medicine. We have now received the enclosed reports from the referees that were asked to re-assess it. As you will see the reviewers are now supportive and I am pleased to inform you that we will be able to accept your manuscript pending a few final editorial amendments.

***** Reviewer's comments *****

Referee #2 (Remarks for Author):

Publish as it is now.

Referee #3 (Comments on Novelty/Model System for Author):

As the title indicates, this approach identifies *potential* therapeutic targets, making these studies largely preclinical. Medical impact will be higher when therapeutic targets identified by this approach are validated in human trials.

Referee #3 (Remarks for Author):

The authors have addressed the reviewers' concerns very thoughtfully and thoroughly.

Corresponding Author Name: Sarah Jurmeister

Manuscript Number: EMM-2017-08274